# Retrospective Validation of a 168-Gene Expression Signature for Glioma Classification on a Single Molecule Counting Platform

**DOI:** 10.3390/cancers13030439

**Published:** 2021-01-25

**Authors:** Paul Minh Huy Tran, Lynn Kim Hoang Tran, Khaled bin Satter, Sharad Purohit, John Nechtman, Diane I. Hopkins, Bruno dos Santos, Roni Bollag, Ravindra Kolhe, Suash Sharma, Jin Xiong She

**Affiliations:** 1Center for Biotechnology and Genomic Medicine, Medical College of Georgia, Augusta University, 1120, 15th St, Augusta, GA 30912, USA; ptran@augusta.edu (P.M.H.T.); lytran@augusta.edu (L.K.H.T.); fbinsatter@augusta.edu (K.b.S.); spurohit@augusta.edu (S.P.); jnechtma@augusta.edu (J.N.); dhopkins@augusta.edu (D.I.H.); bdossantos@augusta.edu (B.d.S.); 2Department of Obstetrics and Gynecology, Medical College of Georgia, Augusta University, 1120, 15th St, Augusta, GA 30912, USA; 3Department of Undergraduate Health Professionals, College of Allied Health Sciences, Augusta University, 1120, 15th St, Augusta, GA 30912, USA; 4Department of Pathology, Medical College of Georgia, Augusta University, 1120 15th St, Augusta, GA 30912, USA; rbollag@augusta.edu (R.B.); rkolhe@augusta.edu (R.K.); susharma@augusta.edu (S.S.)

**Keywords:** brain cancers, gliomas, biomarker, single molecule counting, transcriptomics, retrospective validation

## Abstract

**Simple Summary:**

Molecular classification of cancers has the potential to automate and decrease errors in cancer classification. We previously showed that transcriptomic classification is comparable to methylomic and mutation methods for glioma classification and may provide benefit in predicting survival prognosis. Here we validate the transcriptomic classification method on a single molecule counting gene expression platform using formalin-fixed paraffin embedded samples.

**Abstract:**

Gene expression profiling has been shown to be comparable to other molecular methods for glioma classification. We sought to validate a gene-expression based glioma classification method. Formalin-fixed paraffin embedded tissue and flash frozen tissue collected at the Augusta University (AU) Pathology Department between 2000–2019 were identified and 2 mm cores were taken. The RNA was extracted from these cores after deparaffinization and bead homogenization. One hundred sixty-eight genes were evaluated in the RNA samples on the nCounter instrument. Forty-eight gliomas were classified using a supervised learning algorithm trained by using data from The Cancer Genome Atlas. An ensemble of 1000 linear support vector models classified 30 glioma samples into TP1 with classification confidence of 0.99. Glioma patients in TP1 group have a poorer survival (HR (95% CI) = 4.5 (1.3–15.4), *p* = 0.005) with median survival time of 12.1 months, compared to non-TP1 groups. Network analysis revealed that cell cycle genes play an important role in distinguishing TP1 from non-TP1 cases and that these genes may play an important role in glioma survival. This could be a good clinical pipeline for molecular classification of gliomas.

## 1. Introduction

Gliomas, are neoplasms arising from the cerebral hemisphere [1], which exhibit highly variable responses to chemoradiation therapy and survival prognosis [2,3,4,5]. These tumors are classified using a combination of histology and molecular testing, according to the WHO 2016 classification [1,5]. Histological diagnoses include glioblastoma, astrocytoma, and oligodendroglioma, whereas the use of mixed gliomas such as oligoastrocytoma, as described in the WHO 2007 classification [6], is currently discouraged. Molecular testing [7,8] includes the testing of mutations in the gene isocitrate dehydrogenase (*IDH*) [9]. These mutations in *IDH* genes further refines the histological classification of primary glioblastoma (*IDH* wild type, IDHwt) [10,11] compared to lower grade diffuse gliomas, which includes both diffuse astrocytoma and oligodendroglioma (*IDH* mutated, IDHmut) [5]. Molecular testing of chromosome arm 1p/19q co-deletion status defines oligodendroglioma [10,11,12].

There is a 14% disagreement rate between histology and molecular testing [13]. The inter-observer disagreement with histology diagnosis is also high [12]. The classification of gliomas becomes more complex due to the evolution, understanding, and interpretation of the constantly updated classification literature. It is therefore necessary to develop objective and automated measures of tumor classification, such as through transcriptomic profiling.

Methylation-based classification is another method that automates tumor classification and may identify worse prognosis subgroups [8,14]. Verhaak et al. described the transcriptomic classification of gliomas into proneural, neural, and mesenchymal subtypes. The Cancer Genome Atlas (TCGA) and Capper et al. have individually described methylomic classification systems that are highly similar. While most samples show concordance between IDH-codel and methylomic classification, the methylomic method provides further stratification based on worse prognosis [8,14].

We recently reported a transcriptomics-based classification scheme for gliomas [13]. The Cancer Genome Atlas (TCGA) brain cancer cohort [8] was classified into four-subtypes: transcriptome profile one (TP1), T2a, TP2b and TP3. These subtypes of gliomas were validated in the REpository for Molecular BRAin Neoplasia DaTa (REMBRANDT) cohort [15]. We reported that the TP1 subtype is associated with IDHwt and glioblastoma cases but may contain additional non-glioblastoma glioma cases associated with worse prognosis. Our report provides strong evidence that the transcriptomic platform is also a robust approach to classify brain tumors.

Both methylome and TP based platforms can reduce pathologist time and processing resources for many institutions which use histological analysis. Implementation of TP may be beneficial as an alternative or supplement to methylome and *IDH* classification. RNA profiling can be processed from multiple platforms, such as RNA sequencing, microarray, PCR, single molecule counting, and others, whereas methylation profiling mostly uses microarray platforms in the clinical setting [14]. We selected the Nanostring [16] single molecule counting nCounter^TM^ platform over PCR for its efficiency at validating multi-gene panels and over RNA sequencing for its lower cost for gene panels with lower gene numbers and over both PCR and RNA sequencing platforms for its lower hands-on time, ease of use, and readiness for clinical translation [17]. Moreover, the nCounter platform has the potential for higher sample throughput than the other technologies.

Molecular profiling methods like methylomics and transcriptomics may have limitations as well. Both methods require more tissue than traditional histological methods. Obtaining more tissue may not be feasible in some patients. Additionally, biases may exist in trained algorithms which rely too heavily on one data center or profiling platform.

Pathology archives store and have ready access to formalin fixed paraffin embedded (FFPE) tissues, making this a rich resource for RNA profiling and classification. Storage of flash frozen tissues is becoming common practice due to the increasing popularity of molecular tests. A gene expression quantification platform which can readily accept both sample types would be ideal for clinical applications. The nCounter technology facilitates a rapid, sensitive, and reproducible transcriptomic profile from a complex mixture of RNA obtained from both FFPE and flash frozen tissues. This technique has been successfully applied to define candidate markers for acute myeloid leukemia [18], breast cancer [17,19], and autoimmune diseases [20]. These results suggest single molecule counting is applicable to direct analysis of customized TP signatures in human diseases in clinics [21].

We describe our initial evaluation of nCounter analysis of custom genes in a brain cancer cohort at Augusta University. We present our results on the technical reproducibility of the technology. Our study demonstrates validation of 168 gene classifier [13] developed for classification of gliomas on nCounter platform using retrospectively collected brain cancer samples.

## 2. Results

### 2.1. Specimen Acquisition, RNA Isolation and Quantification

#### 2.1.1. Specimen Acquisition

This was a single-institution, retrospective study examining brain glioma tissue samples from patients seen at Augusta University (AU) Medical Center. The formalin fixed paraffin embedded (FFPE) tissue blocks were obtained from the Department of Pathology. The clinical and demographic information was obtained from the Georgia Cancer Center (GCC) Cancer Registry.

We queried the Cancer Registry for all potential “brain cancers” with a pathology specimen ID. All archived tumors extracted from “Brain” at Augusta University were requested from the Cancer Registry. Only the cases with archived tissue and with a histological diagnosis of “astrocytoma, oligodendroglioma, oligoastrocytoma, or glioblastoma” were kept. We found 296 potential brain tumors archived between 2000 and 2018 under these criteria (Figure 1). This included 289 potential unique FFPE glioma tissues from the Cancer Registry and seven potential unique flash frozen tissues from GCC Biorepository. During tissue retrieval, 76 FFPE tumor blocks were not found and 96 tumors which were “needle biopsies” were excluded, since these blocks with less tissue must be kept for potential clinical use.

We successfully retrieved seven flash frozen glioma tumors and 117 FFPE glioma tissue blocks for RNA isolation. We used 2 mm diameter cores of the FFPE tissue to target areas with high tumor content in the tissue block. These cored locations and histologic classifications were verified by a board-certified pathologist. In total, one block was cored in three separate locations, 49 blocks were cored in two separate locations, and 67 blocks were cored in one location. The seven flash frozen tissues were cored if the tissue was greater than 4 mm in diameter, otherwise all of the tissue was used for RNA extraction.

Of the isolations, 89 RNA specimens from FFPE blocks and seven RNA specimens from flash frozen tissue were run on the nCounter. After quality control (QC) of the RNA, gene quantification data from the nCounter, 34 runs were removed because of errors in positive spike in controls or normalization. On further histologic review by board-certified pathologists, 14 tissues did not meet our inclusion criteria. Forty-eight samples were used for classification, including 41 FFPE and seven flash frozen tissue samples. Thirty-nine samples were successfully classified (Figure 1).

#### 2.1.2. Demographics

The clinical and demographic characteristics of the AU cohort (*n* = 48) are presented in Table 1. The tissue blocks were from 21 Males and 20 Females with median age of 64 years (ranging between 9–79 years). Histologically the brain tumors were classified as astrocytoma (*n* = 10), glioblastoma (*n* = 27), oligoastrocytoma (*n* = 4), and oligodendroglioma (*n* = 7) (Table 1). We applied our transcriptomic classification algorithm [13] to classify these 48 FFPE tissues into TP1 (*n* = 30), TP2A (*n* = 4), TP2B (*n* = 4), and TP3 (*n* = 1).

#### 2.1.3. nCounter replicability

Prior to large-scale analysis, day to day reproducibility of the pipeline was evaluated using seven FFPE samples. Reproducibility was evaluated starting from extraction of RNA to completion of the nCounter analysis. The single molecule count data were used to evaluate the within day and between days reproducibility analysis. We found high day-to-day replicability (average pairwise correlation 0.987) of our gene expression quantification platform from six RNA samples run 48 h apart (Figure 2).

When count data were plotted against the replicates, the data lined up along the regression line with a tight distribution (Figure 2A). We also performed gene expression quantification for multiple cores from the same tumor and found that cores from the same tumor were more similar to each other than cores from other tumors (Figure 2B). The average pairwise correlation between cores from the same tumor was 0.871. Additionally, all cores from the same tumor were classified as the same transcriptome group (Appendix A). Thus, despite the intratumor heterogeneity, our transcriptome classification method is still able to robustly make group calls. Figure 2C is a UMAP of all AU classified samples after integration with TCGA data and it shows that our pipeline can successfully classify both FFPE and flash frozen samples.

### 2.2. Supervised Classification

We previously reported on our ensemble algorithm which combines 1000 linear support vector classifiers trained from TCGA RNASeq data clustered using our combined UMAP and density-based clustering algorithm [13]. We applied this algorithm to our AU cohort data after batch normalization, z-score transformation, and using Empirical Bayes to combine these data with the TCGA data. Empirical Bayes was effective in combining the two data sets (Appendix A). The supervised UMAP approach shows that AU Nanostring cases cluster well into one of the four major glioma transcriptome profiles, regardless of year of tissue source (Figure 2C).

The ensemble model classification using 1000 linear support vector classification models (Appendix A) for the AU samples is 30 in TP1, 4 in TP2a, 4 in TP2b, 1 in TP3, and 9 ambiguous (Figure 3A, Appendix A). The median classification confidence for TP1, TP2a, TP2b, and TP3 were 0.99, 0.75, 0.88, and 0.85, respectively (Appendix A). The 30 TP1 patients include 20 glioblastoma not otherwise specified (NOS), 6 glioblastoma IDHwt, 1 Diffuse astrocytoma NOS, 1 anaplastic astrocytoma NOS, 1 oligodendroglioma NOS and 1 oligodendroglioma IDHmut and 1p/19codel cases. The TP2a patients are 1 oligoastrocytoma NOS, 1 oligodendroglioma NOS and 2 oligodendroglioma IDHmut and 1p/19codel. The TP2b cases are 2 diffuse astrocytoma NOS, 1 anaplastic astrocytoma NOS and 1 anaplastic astrocytoma IDHwt. The only TP3 patient was classified as diffuse astrocytoma IDHmut. All nine ambiguous patients have very low probability of belonging to TP1 but assignment to the non-TP1 groups is less confident. The high rate of ambiguity is likely due to the imprecision of integrating RNAseq with the smaller Nanostring dataset (Appendix A).

To examine survival difference, TP1 cases were compared to non-TP1 cases. As expected, TP1 patients have worse survival than non-TP1 cases (median survival time, 12.1 vs. 83.5 months, likelihood ratio test *p* = 0.005, Figure 3C).

### 2.3. Network Analysis

These 168 genes in the supervised model were enriched for the “G0 and Early G1”, “Cell Cycle Checkpoints”, “G2/M Transition”, and “Transcriptional Regulation by TP53” pathways and that these pathways are upregulated in TP1 compared to other transcriptomic profiles (Figure 4). These findings agree with our previous pathway analysis of TCGA data [13].

## 3. Discussion

We validated a classification method using gene expression profiles in glioma tissues in an institutional cohort using a single-molecule counting platform. While the Nanostring platform has even been used for identifying fusion variants in pediatric gliomas [22,23], it has not been used for glioma classification. We had 79 RNA extractions fail quality control due to low RNA content (Figure 1). Most of these failed RNA extractions started with low tissue quantity. We did not pre-specify a lower tissue volume threshold for RNA extraction but will do this in the future.

Our high technical replicability (Figure 2A) agrees with previous reports of nCounter platform [21,24,25]. Other studies have demonstrated similar replicability of the nCounter platform compared to other gene expression quantification platforms, such as Affymetrix [26], and quantitative real time PCR (qRT-PCR) [27]. qRT-PCR is technically difficult to perform on FFPE tissue because the formalin fragments the nucleic acids and can inhibit enzyme activity [28]. Both these factors limit the activity of reverse transcriptase and DNA polymerase to identify substrates. This is one reason the nCounter platform was developed, since the assay protocol does not depend on any enzymatic activity [16]. Due to (1) the technical challenges of qRT-PCR for FFPE tissue, (2) the extensive validation of the Nanostring platform, (3) the high number of genes of interest, and (4) the differences between count data and the Ct data, we did not validate the Nanostring gene expression data with qRT-PCR.

Although gliomas have high intratumoral heterogeneity [29,30], we demonstrated that different cores from the same tumor have higher correlations to each other than to cores from different tumors (Figure 2B). We also showed that cores from the same tumor are classified as the same tumor subtype (Figure 2C). This analysis shows that tumor subtype prediction is robust to the tumor heterogeneity found in the gliomas we analyzed.

In our previous publication, our algorithm identified non-glioblastoma TP1 samples in the TCGA cohort and that those samples had a survival prognosis similar to glioblastoma cases [13]. This supports the potential for the algorithm to identify cases with glioblastoma-like survival prognosis. Just as in our previous publication. Similarly, in our current study, AU glioma TP1 cases also included several non-glioblastoma samples which likely represent additional samples identified by our algorithm to have worse prognosis like other glioblastomas (Figure 3C).

Gene set enrichment analysis showed that the “cell cycle”, “mitosis”, and “transcriptional regulation by TP53” pathways are significantly enriched in TP1 (Figure 4) compared to non-TP1 cases and these genes are associated with glioma survival prognosis. The role of the TP53 pathway is well established in glioma; mutated TP53 upregulates *MYC*, *EGFR*, *PNCA* and downregulates *p21*, *CD95Fas*, *PTEN*. The cell cycle gene upregulation may originate from cancer cells or immune cells, which are prevalent in glioblastoma [13]. The presence of cell cycle and TP53 pathways enriched in TP1, both of which are established pathways associated with glioblastoma suggests the former [31].

We identified potential drug targets for TP1 cases based on our pathway analysis. Most of the potential drug targets are in the cell cycle pathway, including microtubule inhibitors, polo-like kinase (PLK) inhibitors, and cyclin-dependent kinase inhibitors [32]. Of the microtubule inhibitors, docetaxel [33,34], epothilone B [35,36], ixabepilone [36], and sagopilone [37] have been tested in Phase II clinical trials. Of the PLK inhibitors, GW843682X [38] and JNJ-10198409 [39] have been tested in pre-clinical models. Of the CDK inhibitors, abemaciclib has been tested in phase I trials [40,41], and palbociclib has been tested in pre-clinical models [40,42,43].

One limitation of our study is our incomplete *IDH* mutation and 1p/19q codeletion data (Figure 3A). Thus, our classification could only be compared to histology. Since many of these samples were collected before the discovery of the *IDH* mutation [9,10,11], this information was not available from the pathology report. Additionally, due to the high degree of nucleic acid fragmentation in the FFPE samples, we could not sequence to determine *IDH* mutation status.

The validity of our transcriptome ensemble classifier was demonstrated using gene expression data from the Nanostring platform. Although the Nanostring dataset is small, it does indicate that the classifier can be assayed by a technology that is easy to implement clinically. We report a tumor classification strategy robust to tissue storage condition (flash frozen vs. FFPE) and comparable to gene expression quantification techniques (RNASeq, microarray, and Nanostring). This versatility provides an advantage over methylomic classification approaches which relies on a microarray-based platform. These data taken together make the Nanostring platform a strong candidate for classification and prognostication for gliomas [26,44,45,46].

## 4. Materials and Methods

### 4.1. Augusta University Sample Collection and Processing

#### 4.1.1. Study Participants

We collected 34 formalin-fixed paraffin embedded (FFPE) tissue blocks from the Augusta University Medical Center Department of Pathology from specimens archived between 2002 and 2018. We obtained 7 flash frozen brain cancer tissue from the Georgia Cancer Center Tumor Tissue and Serum Biorepository. Inclusion criteria included histologic diagnosis of oligoastrocytoma, astrocytoma, oligodendroglioma, anaplastic oligodendroglioma, or glioblastoma. Specimens were excluded if they were biopsy samples. No statistical methods were used to predetermine sample size. The investigators were blinded to allocation during experiments and outcome assessment.

Demographic and clinical data were collected through the AU Cancer Registry and validated through the patients’ electronic health records. Data and sample collection were conducted through a consent-waived retrospective arm of an Institutional Review Board approved study (Biomarkers and Therapeutics in Cancer). The study was conducted according to the Declaration of Helsinki (1996) and was approved by the institutional review board at the Augusta University. Overall-survival was used as the clinical endpoint for the analysis of the data.

#### 4.1.2. Specimen Characteristics and Assay Methods

For all FFPE tissue blocks collected, a board-certified pathologist reviewed the corresponding H&E stained tissue section, classified the tissue according to the WHO guidelines, and identified regions with >60% tumor nuclei. A 2 mm diameter core was then punched from the identified region of the tissue block and dispensed in a 96-well 2D matrix barcode system for sample storage and tracking.

#### 4.1.3. Pipeline for FFPE Tissue RNA Extraction

We developed a customized pipeline for processing of FFPE cores onto the nCounter system. Cores and flash frozen tissue were mechanically and chemically disrupted using a bead homogenizer and Citrisolve (Fisher Scientific, Piscataway, NJ, USA) to deparaffinize and disrupt the FFPE tissues. RNA was then extracted from the lysate using a column-based RNA extraction kit (RNEasy, Qiagen, Hilden, Germany). Qualtiy of isolated RNA was assessed with a Tapestation 2000 and Nanodrop. Concentration of RNA was assessed using bioanalyzer for RNA fragments greater than 300 bp in length. The RNA concentration used for loading amount on nCounter was the lower RNA concentration between the Tapestation calculation and the Nanodrop calculation. RNA with concentration less than 20 ng/uL was not used for downstream analysis. A total of 200 ng RNA were hybridized and loaded on the nCounter.

### 4.2. Statistical Methods

All statistical analyses were performed using the R language and environment for statistical computing (R version 3.5.1; R Foundation for Statistical Computing [47]).

#### 4.2.1. Gene Expression Quantification

We submitted 168 genes (including 8 housekeeping genes) to Nanostring Technologies [16] to develop a Custom Code Set gene expression assay. We hybridized 100–200 ng of RNA per sample (5 μL) reporter probe and capture probe mix from the Custom Code Set according to the manufacturer’s protocol, then purified the target/probe complexes and immobilized them on the NanoString cartridge for data collection using the nCounter Prep Station. Transcript counts were determined using the nCounter Digital Analyzer and outputted in reporter code count (RCC) files. RCC files containing raw transcript counts from each cartridge were analyzed using the nSolver analysis software for quality control (QC) purposes. The software was used to check for imaging, binding, and positive spike-in quality.

The output files from nSolver were read into R for further QC, normalization, and data processing. We normalized the captured transcript counts using multiplicative normalization factors calculated with geometric means to first the codeset’s internal positive controls and then the geometric mean of the reference genes included in our assay (GAPDH, HNRNPL, IPO8, MRPL19, MRPL30, NRF1, RNF10, and TBP). The fully normalized counts were log2-transformed.

#### 4.2.2. TCGA Dataset

TCGA Glioma gene expression data, which contains both RNAseq and gene expression microarray data combined through Empirical Bayes, was downloaded from Ceccarelli et al. 2016. The final dataset contained 1032 samples and 12,717 genes. TCGA Glioma data were centered and scaled. Clinical data were downloaded from the same source and matched to the processed TCGA Glioma data.

Data from Nanostring and TCGA RNAseq were integrated using the Empirical Bayes-based Combat algorithm [48] implemented in the “SVA” package.

#### 4.2.3. Ensemble Transcriptomic Classification (ETC) Algorithm

One hundred and sixty-eight genes were identified for use in supervised classification using two complementary methods. First, significantly differentially expressed genes amongst our four groups were identified through linear models for microarray data (LIMMA) analysis [49] and 26 genes were manually selected based on relevance to brain cancer. The remaining 142 genes were selected using recursive feature elimination with a support vector classifier. These genes were divided into six groups based on the expression differences among the subtypes. For each supervised model, half of the genes in each of six gene groups were randomly selected and then recursive feature elimination was applied removing five genes per iteration until optimal accuracy is reached with the minimal number of genes using “sci-kitlearn” [50]. This was repeated 1000 times, resulting in a dictionary with 1000 entries each with between 29 and 79 genes of the 168 genes. One thousand linear support vector classifiers (LSVC) were developed from the dictionary and the mean accuracy from three-fold cross validation was used to remove any models with an average accuracy less than 95%. All 1000 models passed this step and average model accuracy was 97.6%.

In order to decrease the potential of overfitting, data were split into four folds, where three folds were trained on the unsupervised model classes and the supervised models predicted on the remaining fold. In this way, no sample was used for both training and making classification calls. This results in calls for each sample from 1000 linear SVC models. Models were combined into one ensemble model using a plurality voting method which reports the most popular class and the proportion of the 1000 LSVC models which agree on this most popular class. A confidence score is calculated by taking the proportion of models classifying samples into the most popular class divided by the proportion of models classifying samples into the second most popular class. If the confidence score is greater than 3, then the ensemble model classifies the sample into the most popular class. If the confidence score is less than or equal to 3, than the ensemble model prediction is “ambiguous”.

#### 4.2.4. Survival Analysis

We modeled over-all survival with Kaplan–Meier and Cox proportional hazards and tested for significance with the log rank test all using the “survival” R package. Kaplan-Meier survival plots were made through the “survminer” package. The statistical significance of differences was set at *p* < 0.05, all *p* values were two sided. Patients with no history of recurrence or death were censored at the date of last follow-up visit. Patients who died of natural causes unrelated to cancer were censored at time of death.

#### 4.2.5. Network Analysis

We performed LIMMA [49] to identify the differentially expressed genes between the TP groups in the Nanostring dataset and fitted the model based on the contrast matrix and applied the fold change and significance testing from this model to network package ReactomePA [51], parameters include gene list, and fold change.

## 5. Conclusions

We describe a pipeline for RNA isolation and classification of glioma tumor tissue from formalin fixed and flash frozen tissue. We describe a machine learning algorithm that can classify glioma gene expression data from at least three different platforms (RNASeq, microarray, and nanostring) into a clinically and prognostically relevant subtype. Thus, regardless of the RNA profiling platform, our method can predict glioma subtype. Network analysis indicates cell cycle and cell proliferation pathways represent a key difference between TP1 groups compared to the others.

## Figures and Tables

**Figure 1 cancers-13-00439-f001:**
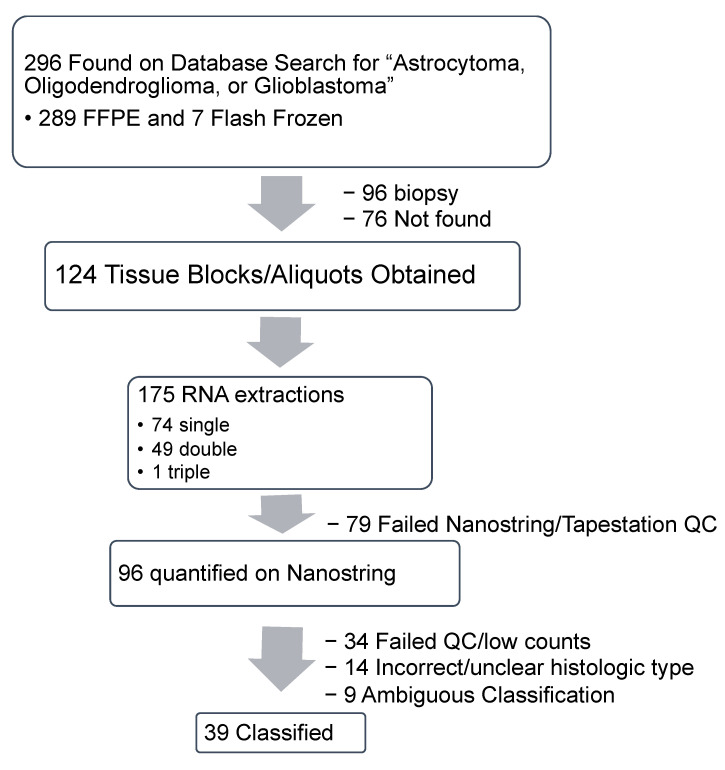
Retrospective study design and study flow chart. FFPE: formalin fixed paraffin embedded; QC: quality control.

**Figure 2 cancers-13-00439-f002:**
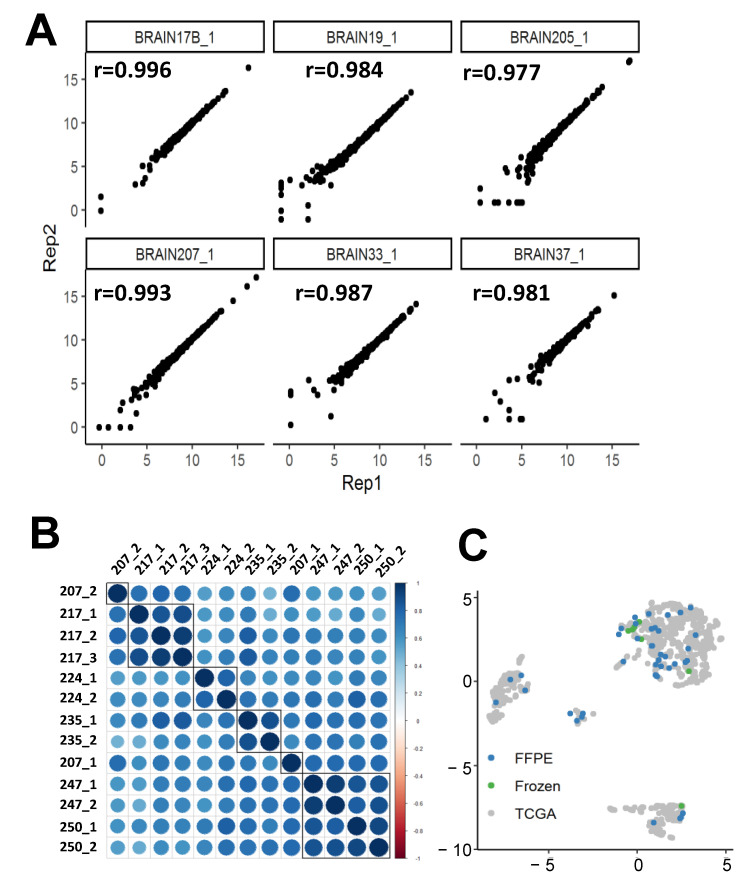
Robustness and Replicability of single molecule counting nCounter platform from Nanostring. (**A**) Day-to-Day replicability of nCounter runs (**B**) Clustered Correlation Heatmap of nCounter count data for 13 unique FFPE cores from 6 glioma surgical cases. (**C**). Uniform Manifold Approximation and Projection (UMAP) plot of combined the Cancer Genome Atlas (TCGA) and AU data. Tissue type of AU samples (green and blue) and TCGA samples (grey).

**Figure 3 cancers-13-00439-f003:**
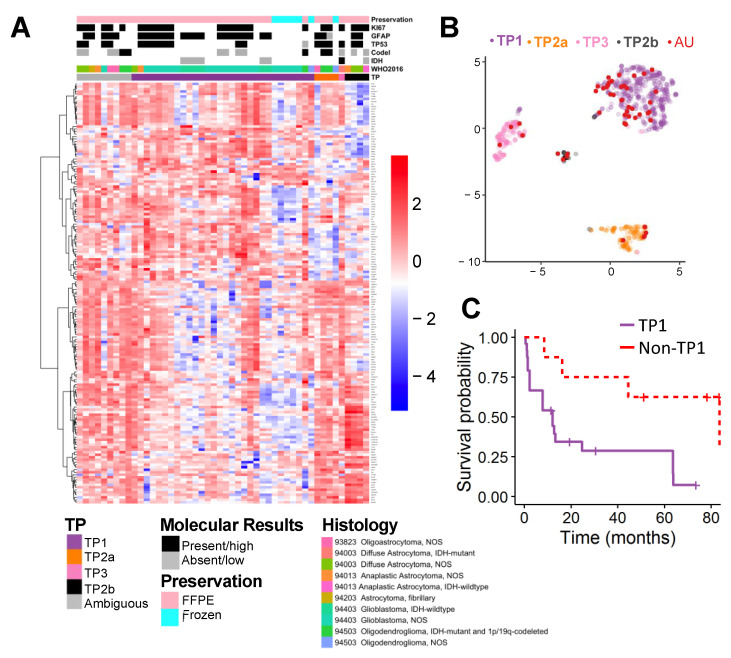
Molecular distribution of Augusta University (AU) cohort samples and survival analysis. (**A**) Heatmap of AU Nanostring classification results, molecular characteristics, and clinical characteristics sorted based on supervised transcriptome profile classification. (**B**) UMAP representation of integrated TCGA data with AU cases projected (red). Colors show supervised transcriptome profiles. (**C**) Kaplan–Meier survival estimates for cases classified as TP1 (purple) compared to non-TP1 (HR 4.5 95% CI 1.3–15.4, median survival time, 12.1 vs. 83.5 months, likelihood ratio test, *p* = 0.005).

**Figure 4 cancers-13-00439-f004:**
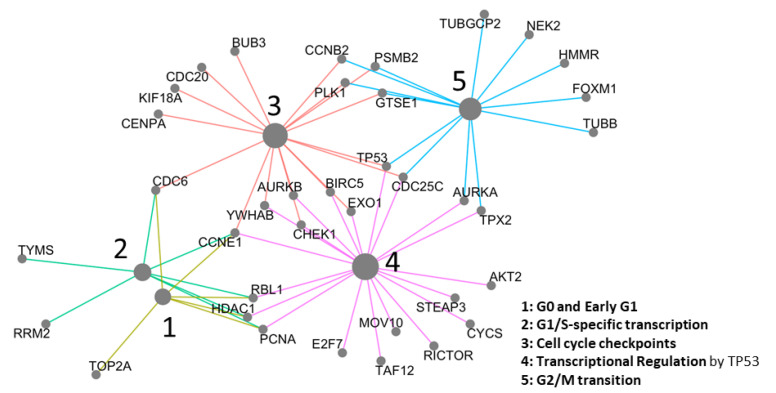
Network analysis of the 168 Supervised Model genes.

**Table 1 cancers-13-00439-t001:** Demographics and clinical data for the samples collected at Augusta University (AU).

Clinical Variable	Number of Subjects
Age (Years, median (range))	64 (9–79)
Year specimen collected	2003–2016
Sex	
Male	21
Female	20
Unknown	7
Race	
White	34
AA	7
Unknown	7
WHO 2016 Classification	
Oligoastrocytoma, NOS	3
Diffuse Astrocytoma, *IDH*-mutant	1
Diffuse Astrocytoma, NOS	5
Anaplastic Astrocytoma, NOS	3
Anaplastic Astrocytoma, *IDH*-wildtype	1
Astrocytoma, fibrillary	1
Glioblastoma, *IDH*-wildtype	6
Glioblastoma, NOS	21
Oligodendroglioma, *IDH*-mutant and 1p/19q-codeleted	5
Oligodendroglioma, NOS	2
Transcriptome Profile	
TP1	30
TP2	4
TP3	1
TP4	4
Ambiguous	9
Preservation	
FFPE ^1^	41
Frozen	7
*IDH1*	
Mutated	1
WT	7
Unknown	40
1p/19q Co-deletion	
Present	3
WT	11
Unknown	34

^1^ Formalin-Fixed Paraffin Embedded. IDH: isocitrate dehydrogenase; NOS: not otherwise specified.

## Data Availability

The data presented in this study are available on request from the corresponding author. The data are not publicly available due to privacy and ethical reasons.

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
