# Peer review of "Retrospective Validation of a 168-Gene Expression Signature for Glioma Classification on a Single Molecule Counting Platform"

_cancers, 2021, doi:10.3390/cancers13030439_

Round 1
Reviewer 1 Report
General comments, title, and abstract
- The title is concise and appropriate to reflect the major theme of the paper.
- All abbreviations should be revised and defined at their first use. In the abstract, for example, authors used “AU” to refer to Augusta University without defining it at its first use. Same applies to TP which refers to transcriptomic profile.
- Since nothing is definite, I suggest replacing ‘should’ with ‘could’ in: “This should be a good clinical pipeline.”
- Abstract can be improved. More details on the methodology of the protocol could be added.
- The manuscript could benefit from minor editing for grammar, missing words, and subject-verb agreement, etc. It is recommended that authors delete irrelevant “general” phrases and sentences, repeated and unneeded words. They should use short sentences.
Introduction
- Although authors mentioned advantages of methylome and TP-based platforms in “reducing pathologist time and processing resources,” it would be also beneficial to state disadvantages or hitches and limitations of using this technology.
- Authors are advised to elaborate more on the molecular background of glioma as an introduction to the topic.
Methods and Results:
- Please correct the typo in the sentence: “patients have worse survival than non-TP1 cases (median survival time, 12.1 vs 83.5 months, LRT p = 0.005, Figure 5C).” Figure 3C instead of 5C.
- Authors presented the network analysis of the 168 Supervised Model genes without further analysis of their clinical significance and their pharmacogenetic importance. I think it is of utmost importance to identify drug targets based on molecular data extracted from the protocol used in the manuscript.
- Authors developed a protocol for RNA extraction and quality control from 2mm cores of FFPE and flash frozen tissue, and further applied this protocol and obtained gene expression data of 168 genes for 41 unique glioma samples. However, no further analysis was performed to validate it.
Discussion:
- “TP1 also included several non-glioblastoma samples which likely represent additional samples identified by our algorithm to have worse prognosis like other glioblastomas.” This questions the specificity of the protocol generated and algorithm established.
- “Transcriptional regulation by TP53 pathways are significantly enriched in TP1 (Fig 4) compared to non-TP1 cases and these genes are associated with glioma survival prognosis. The role of the TP53 pathway is well established in glioma; mutated TP53 upregulates MYC, EGFR, PNCA and downregulates p21, CD95Fas, PTEN etc.” Authors were very general in presenting their data. More validation via qRT-PCR should have been done on the cohort of patients included.
- Discussion section is poorly written. Authors should focus on the main findings and avoid repeating results presentation in the discussion. Authors should also correlate their findings with what has been published in literature. Clinical relevance should be added.
Author Response
We thank the reviewers for their comments and suggestions. We feel they improve the scientific content as well as the communication of our research findings. We have rewritten major parts of the introduction and discussion as requested by the reviewers.
Reviewer 1
General comments, title, and abstract
- The title is concise and appropriate to reflect the major theme of the paper.
Thank you
- All abbreviations should be revised and defined at their first use. In the abstract, for example, authors used “AU” to refer to Augusta University without defining it at its first use. Same applies to TP which refers to transcriptomic profile.
We appreciate the reviewer for noting this and have edited the manuscript for all abbreviation issues.
- Since nothing is definite, I suggest replacing ‘should’ with ‘could’ in: “This should be a good clinical pipeline.”
We made the change as suggested.
- Abstract can be improved. More details on the methodology of the protocol could be added.
We have added details of the tissue curation, RNA extraction, RNA quantification, and subtype prediction to the abstract as requested.
- The manuscript could benefit from minor editing for grammar, missing words, and subject-verb agreement, etc. It is recommended that authors delete irrelevant “general” phrases and sentences, repeated and unneeded words. They should use short sentences.
We thank the reviewer for this comment and have edited the manuscript by using simple, non-general phrases and editing grammatical errors as requested.
Introduction
- Although authors mentioned advantages of methylome and TP-based platforms in “reducing pathologist time and processing resources,” it would be also beneficial to state disadvantages or hitches and limitations of using this technology.
We have included a section detailing potential problems with tissue processing and algorithm bias when using the methylome and TP-based platforms in the introduction as requested.
- Authors are advised to elaborate more on the molecular background of glioma as an introduction to the topic.
This is an excellent idea and we have written more on the known molecular background of gliomas.
Methods and Results:
- Please correct the typo in the sentence: “patients have worse survival than non-TP1 cases (median survival time, 12.1 vs 83.5 months, LRT p = 0.005, Figure 5C).” Figure 3C instead of 5C.
Thank you for noting this. We have corrected this typo.
- Authors presented the network analysis of the 168 Supervised Model genes without further analysis of their clinical significance and their pharmacogenetic importance. I think it is of utmost importance to identify drug targets based on molecular data extracted from the protocol used in the manuscript.
This is an excellent point and we have added this section to the manuscript.
- Authors developed a protocol for RNA extraction and quality control from 2mm cores of FFPE and flash frozen tissue, and further applied this protocol and obtained gene expression data of 168 genes for 41 unique glioma samples. However, no further analysis was performed to validate it.
Thank you for making this point. We have further explained in the manuscript that the core technology used in this project, the Nanostring platform, has been extensively validated in the literature and further that our sample type is more amenable to analysis by this platform compared to RNASeq or qRT-PCR. Please see the discussion for more details.
Discussion:
- “TP1 also included several non-glioblastoma samples which likely represent additional samples identified by our algorithm to have worse prognosis like other glioblastomas.” This questions the specificity of the protocol generated and algorithm established.
Thank you for pointing this out. We have further clarified that our algorithm also identified non-glioblastoma TP1 samples in the TCGA cohort and that those samples had survival prognosis similar to glioblastoma cases. Supporting the potential for algorithm to identify cases with glioblastoma-like survival prognosis. Please see the discussion for more details.
- “Transcriptional regulation by TP53 pathways are significantly enriched in TP1 (Fig 4) compared to non-TP1 cases and these genes are associated with glioma survival prognosis. The role of the TP53 pathway is well established in glioma; mutated TP53 upregulates MYC, EGFR, PNCA and downregulates p21, CD95Fas, PTEN etc.” Authors were very general in presenting their data. More validation via qRT-PCR should have been done on the cohort of patients included.
Thank you for this comment. We further detail in our manuscript the technical difficulty of performing qRT-PCR on FFPE samples. We explain that this is one reason the Nanostring platform was developed. Please see the discussion for more details.
- Discussion section is poorly written. Authors should focus on the main findings and avoid repeating results presentation in the discussion. Authors should also correlate their findings with what has been published in literature. Clinical relevance should be added.
We have extensively edited the discussion as suggested by the reviewer. We removed all sections redundant to the results and added sections to relate our findings to published literature as well as clinical relevance.
Reviewer 2 Report
Minh Huy Tran et al validated a transcriptomic classification method developed in their lab using the Nanostring platform. They showed that only a limited number of genes is sufficient to succesfully classifiy gliomas based on their transcriptomic profile. However, there are some issues that need to be addressed:
- RNA quality seems to be an issue when using this algorithm/platform. From 175 RNA extractions, 113 failed the quality control steps this does not seem to be very reasonable for daily practice. Please elaborate.
- Do I understand correctly that the histology was not reviewed again before included in the study? Why not? A lot has changed regarding tumor classification since 2000.
- IDH1 status and 1p/19q status were unknown in the majority of cases which makes a reasonable integrated histomolecular diagnosis impossible according to current guidelines. Please consider re-classifying tumor diagnosis.
- As presented in figure 3 the most important overlap between histology and transcription class is glioblatoma and TP1. This does not at all seem surprising. However, the correlation with the other subtypes is rather poor. Most of the time glioblastoma is not a difficult diagnosis but the differentiation between IDH wt or mutant very important for the prognosis and survival. I do not see this reflected within the currently presented data.
- The network analysis is in line with the previous statement. If glioblastoma is most robustly classified the activated pathways show manly overlap with genes involved in gloiblastoma tumorigenisis.
Author Response
We thank the reviewers for their comments and suggestions. We feel they improve the scientific content as well as the communication of our research findings. We have rewritten major parts of the introduction and discussion as requested by the reviewers.
- RNA quality seems to be an issue when using this algorithm/platform. From 175 RNA extractions, 113 failed the quality control steps this does not seem to be very reasonable for daily practice. Please elaborate.
Thank you for noting this. Many of our samples were biopsies and very minimal tissue could be recovered for RNA isolation. This is likely the reason for the high QC failure rate. In the future, we could specify a lower limit for tissue accepted for RNA isolation. We now detail this in the discussion.
- Do I understand correctly that the histology was not reviewed again before included in the study? Why not? A lot has changed regarding tumor classification since 2000.
Histology was reviewed by a board-certified pathologist in 2019 for all blocks used in the study, we have added this to the methods.
- IDH1 status and 1p/19q status were unknown in the majority of cases which makes a reasonable integrated histomolecular diagnosis impossible according to current guidelines. Please consider re-classifying tumor diagnosis.
We agree with the reviewer and have highlighted this limitation in the discussion. We clarify that our diagnosis is only based on histological characteristics and so does not match the WHO 2016 guidelines for glioma histomolecular classification
- As presented in figure 3 the most important overlap between histology and transcription class is glioblatoma and TP1. This does not at all seem surprising. However, the correlation with the other subtypes is rather poor. Most of the time glioblastoma is not a difficult diagnosis but the differentiation between IDH wt or mutant very important for the prognosis and survival. I do not see this reflected within the currently presented data.
We agree with the reviewer. Unfortunately, our attempts to sequence the samples for IDH mutation status have not been successful thus far. This is why we used histology only for comparison. We have added this limitation to the discussion.
- The network analysis is in line with the previous statement. If glioblastoma is most robustly classified the activated pathways show manly overlap with genes involved in gloiblastoma tumorigenisis.
We agree and detail more in discussion as stated above. We also discussed the importance of identifying non-glioblastoma TP1 cases for our study since our previous publication found that our classification method could identify non-glioblastoma TP1 cases with survival prognosis similar to glioblastoma cases.
Round 2
Reviewer 1 Report
Thank you for addressing all my comments.
Author Response
Thank you for reviewing this paper! We have spell checked the paper as well.
Reviewer 2 Report
The authors succesfully discussed the limitations of their study. However, I still feel that classification of gliomas should be made according to the WHO 2016 guidelines and not solely based on histology.
Author Response
We thank the reviewer for this feedback and have edited the manuscript as requested. Our pathologist has worked through each of our patient samples and updated the patient to the WHO2016 guidelines for all patients. We have provided this data as a new column in our supplemental table. We have updated table 1, figure 3, and our results section to reflect the changes in classification based on the WHO2016 guidelines.
Round 3
Reviewer 2 Report
no further comments